# Time-Dependent Internalization of Polymer-Coated Silica Nanoparticles in Brain Endothelial Cells and Morphological and Functional Effects on the Blood-Brain Barrier

**DOI:** 10.3390/ijms22041657

**Published:** 2021-02-06

**Authors:** Aniela Bittner, Fabien Gosselet, Emmanuel Sevin, Lucie Dehouck, Angélique D. Ducray, Véronique Gaschen, Michael H. Stoffel, Hansang Cho, Meike Mevissen

**Affiliations:** 1Division of Veterinary Pharmacology and Toxicology, Vetsuisse Faculty, University of Bern, Länggassstrasse 124, 3012 Bern, Switzerland; aniela.bittner@vetsuisse.unibe.ch (A.B.); angelique.ducray@vetsuisse.unibe.ch (A.D.D.); 2Blood-Brain-Barrier Laboratory, University of Artois, UR265, Faculté Jean Perrin, Rue Jean Souvraz–SP 18, 62307 Lens, France; fabien.gosselet@univ-artois.fr (F.G.); emmanuel.sevin@univ-artois.fr (E.S.); lucie.dehouck@univ-artois.fr (L.D.); 3Division of Veterinary Anatomy, Vetsuisse Faculty, University of Bern, Länggassstrasse 120, 3012 Bern, Switzerland; veronique.gaschen@vetsuisse.unibe.ch (V.G.); michael.stoffel@vetsuisse.unibe.ch (M.H.S.); 4Institute of Quantum Biophysics, Department of Biophysics, Department of Intelligent Precision Healthcare Concergence, Sungkyunkwan University, 2066 Seobu-ro, Jangan-gu, #868715 N-Center Suwon-si, Gyeonggi-do 16419, Korea; h.cho@g.skku.edu

**Keywords:** co-culture, 3D model, permeability, transendothelial electrical resistance

## Abstract

Nanoparticle (NP)-assisted procedures including laser tissue soldering (LTS) offer advantages compared to conventional microsuturing, especially in the brain. In this study, effects of polymer-coated silica NPs used in LTS were investigated in human brain endothelial cells (ECs) and blood-brain barrier models. In the co-culture setting with ECs and pericytes, only the cell type directly exposed to NPs displayed a time-dependent internalization. No transfer of NPs between the two cell types was observed. Cell viability was decreased relatively to NP exposure duration and concentration. Protein expression of the nuclear factor ĸ-light-chain-enhancer of activated B cells and various endothelial adhesion molecules indicated no initiation of inflammation or activation of ECs after NP exposure. Differentiation of CD34+ ECs into brain-like ECs co-cultured with pericytes, blood-brain barrier (BBB) characteristics were obtained. The established endothelial layer reduced the passage of integrity tracer molecules. NP exposure did not result in alterations of junctional proteins, BBB formation or its integrity. In a 3-dimensional setup with an endothelial tube formation and tight junctions, barrier formation was not disrupted by the NPs and NPs do not seem to cross the blood-brain barrier. Our findings suggest that these polymer-coated silica NPs do not damage the BBB.

## 1. Introduction

Over the past few decades, the use of nanomaterials has increased tremendously and the areas of application have broadened greatly, ranging from nutrition to appliances and electronics to the automobile industry [1]. Nanotechnology is also of great interest in the field of medicine, having resulted in the rise of nanomedicine [2] including diagnostics and therapy such as in targeted drug delivery or in biodegradable scaffolds in the context of laser tissue soldering (LTS) [3]. Injuries of hollow organs, for example, ruptured cerebrovascular aneurysms, can alternatively be treated by LTS which provides faster wound healing and recovery, immediate water tightness and overall reduced procedure time compared to conventional microsuturing [4,5,6]. Hereby, albumin that is trapped in a biodegradable scaffold will be denatured, resulting in tissue fusion. Denaturation is achieved by the transduction of applied laser light into heat by indocyanine green (ICG) [7]. As ICG is photosensitive and unstable in aqueous solution, encapsulation in nanoparticles (NPs) can contribute to the chromophore’s stabilization [8,9].

When applying LTS to brain vessels, NPs will come into direct contact with brain endothelial cells (ECs) after release from the biodegradable scaffold [10]. ECs of the brain microvasculature display tight junctions, strongly interacting with adherens and gap junctions, as well as an absence of fenestrations and a very low pinocytotic activity. These properties contribute to form a blood-brain barrier (BBB) that restricts access to the brain tissue [11,12]. Crosstalk between the BBB endothelial cells and other neural and mural cell types such as pericytes, neurons or astrocytes forms the neurovascular unit (NVU). A potential risk relates to BBB breakdown as a result of the exposure to NPs, thus facilitating the entry of harmful xenobiotics and microorganisms into the brain [13,14,15].

Previously, we investigated the effects of silica-ICG/poly(ε-caprolactone) (PCL) and silica-ICG/poly(ε-caprolactone-poly(l-lactide)) (PLLA) NPs on the immortalized rat brain capillary endothelial cell line rBCEC4. Both types of NPs were taken up into the ECs in a time-dependent manner and to a high extent. About 87% and 84% of rBCEC4 cells had internalized PCL- or PLLA-NPs after 24 h of NP exposure, respectively [16]. Polyisoprene NPs were found to be taken up by the immortalized human ECs hCMEC/D3 without causing an effect on cell viability up to concentrations of 100 µg/mL [17]. Similarly, neither chitosan, nor poly(lactic-co-glycolic acid) (PLGA), nor cationically modified poly-lactide NPs nor gold or quantum rods affected cell viability of brain microvascular ECs (BMVECs) [18]. Contrarily, after uptake into immortalized murine cerebral ECs (bEnd.3) and murine brain astrocyte-like cells, silver (Ag) NPs and TiO_2_-NPs led to decreased cell viability [19].

rBCEC4 cells were shown to express tight junction proteins such as claudin 3, occludin and zonula occludens 1 (ZO-1) and exhibited transendothelial electrical resistance (TEER) values of > 40 Ω∙cm^2^. The paracellular permeability to 4.4 kDa dextran and 70 kDa dextran, given as the apparent permeability coefficient (Papp), was around 1.5 × 10^−5^ cm/sec and 0.5 × 10^−5^ cm/sec, respectively. None of these factors were affected by the exposure to PCL-NPs [16]. The integrity of the BBB, as assessed by TEER and permeability to Lucifer yellow (LY) and dextran, remained unaffected after exposure of hCMEC/D3 cells to polyisoprene NPs [17]. Similar findings were reported after exposure of BMVECs to chitosan, PLGA, cationically modified poly-lactide NPs or gold or quantum rods. None of these types of NPs altered the expression of junctional proteins of the BBB such as claudin 5, occludin, ZO-1 or junctional adhesion molecule 2 (JAM-2) or the integrity of the barrier [18]. Contrary to this, a reduction in TEER and a disruption of the junctional markers claudin 5 and ZO-1 was seen after uptake into bEnd.3 cells and murine brain astrocyte-like cells [19]. Non-modified or surface-modified PLGA NPs led to a concentration-dependent increase in BBB permeability as well [20].

Upon contact to pro-inflammatory stimuli, ECs will become activated, resulting in a release of inflammatory mediators and the up-regulation of adhesion molecules that are needed for recruitment and barrier crossing of leukocytes in the blood. These adhesion molecules include selectins, the intercellular adhesion molecule 1 (ICAM-1) and the vascular cell adhesion molecule 1 (VCAM-1). It has been shown that different types of NPs cause increased expression of these molecules in human umbilical vein ECs (HUVECs) [21] which is mediated by the activation of the NF-κB pathway [22].

The rBCEC4 cell model is rather simplistic as it only included one cell type, the brain ECs. In addition, species differences of brain endothelial cells have been reported [23]. Hence, to get closer to the in vivo situation and to be able to draw more reliable conclusions, the potential effects of PCL- and PLLA-NPs were investigated in a co-culture model involving primary human endothelial cells derived from hematopoietic stem cells (hBLECs) and brain pericytes [24]. Furthermore, most recent research in the area of nanotechnology mainly focused on the use of NPs for targeted drug delivery, investigating the efficiency of drug transport across the barrier. Studies assessing potential adverse effects of NPs on brain ECs or the BBB are sparse. Hence, in this study cell viability, NP uptake, possible induction of inflammation in the ECs and above all interactions of NPs with the BBB were examined. As in vivo vessels exhibit a 3-dimensional (3D) tube-like structure, the co-culture model was transferred to a 3D microfluidic device to allow for vessel formation.

## 2. Results

### 2.1. Effects of Nanoparticles on Cell Viability (MTT) and Cytotoxicity (PI)

CD34+ ECs and pericytes were exposed to three different concentrations of PCL- or PLLA-NPs, ranging from 2.49 × 10^−7^ μg/mL to 24.9 µg/mL, for 2 and 24 h. As depicted in Figure 1, no significant effects were seen after exposure to the two lower concentrations regardless of the duration or NP type. Both PCL- and PLLA-NPs significantly decreased the cell viability after 2 and 24 h with exposure to PCL-NPs resulting in a more pronounced reduction (24 h exposure: 54% or 59% of viable cells after PCL-NP exposure (Figure 1a,c) compared to 69% or 64% after PLLA-NP exposure (Figure 1b,d) of CD34+ ECs or pericytes, respectively) (Figure 1). 

Due to the slight decrease in cell viability, the medium concentration (2.49 × 10^−3^ μg/mL) was chosen for assessment of possible long-term effects of NPs on the cell viability as well as a potential cytotoxic effect on CD34+ ECs and pericytes. PCL-NPs were used to assess cytotoxicity. Both cell types were exposed to PCL-NPs and stimulated with staurosporine as positive control on day in vitro 1 (DIV1). Cell viability and cytotoxicity were measured directly after on DIV1 in a first set of cells. A second set of cells was maintained in culture until DIV4 on which cell viability and cytotoxicity were detected after stimulation with staurosporine. No significant differences between non-exposed controls and PCL-NP-treated cells could be seen in any of the cell types investigated as well as on DIV1 and DIV4 (Figure 2). In addition, Figure A1 shows the Hoechst and PI stainings.

### 2.2. Nanoparticle Uptake in CD34+ ECs and hBLECs

The uptake of PCL- and PLLA-NPs into hBLECs was assessed by transmission electron microscopy (TEM) (Figure 3). Both types of NPs were taken up into hBLECs after 24 h of exposure to concentrations of (24.9 µg/mL). PCL- or PLLA-NPs could be detected mostly in clusters of different sizes (around 80–150, 150–500 and 500–1000 nm). They were present freely in the cytoplasm of the cell or encapsulated in membrane-bound vesicles as shown in Figure 3a,b. Due to limitations of the technique used for high-content analysis, it was not possible to quantify the uptake of NPs in hBLECs as the cells were grown on filters. Instead, quantification was carried out in monoculture of CD34+ ECs (Figure 3c) by means of the fully automated inverted epifluorescence INCell Analyzer. The cells were exposed to PCL- or PLLA-NPs for 0.5, 2 or 24 h at a concentration of 24.9 µg/mL. A time-dependent highly significant increase in the number of cells having internalized NPs was found.

PLLA-NPs seemed to have been taken up to a slightly higher extent. About 83% or 87% of CD34+ ECs had internalized PCL- or PLLA-NPs, respectively, after a 24-h exposure. After NP exposure of hBLECs in the apical compartment, being part of a co-culture, no NPs were found in the pericytes. On the other hand, pericytes in monocultures internalized both types of NPs (Figure A2). However, due to high cell densities in the wells, quantification was not possible. If pericytes only were exposed to PCL-NPs in the basolateral compartment of the co-cultures, no NPs were detected in the hBLECs.

### 2.3. Effects of Nanoparticles on Endothelial Adhesion Molecules and Inflammation

Possible induction of inflammation with activation of ECs was assessed by the expression of the marker NF-κB, p-NF-κB and of several adhesion molecules by means of IF. Both in unexposed controls and PLLA-exposed hBLECs, NF-κB (Figure 4a,c) and p-NF-κB (Figure 4d,f) was shown to be localized to the cytoplasm of the cells and its fluorescence intensity was not altered by the exposure to polymer-coated NPs. Incubation with LPS, as a positive control, led to translocation of the signal to the nucleus (Figure 4b,e).

The adhesion molecules ICAM-1 (Figure 5a–c) and VCAM-1 (Figure 5j–l) were both barely expressed and to a similar extent in untreated control cells and PLLA-exposed cells. Tumor necrosis factor (TNF)-α and Interferon (IFN)-γ stimulation as a positive control resulted in up-regulation of fluorescence intensity by eye of ICAM-1 (Figure 5b) and VCAM-1 (Figure 5k). Furthermore, ICAM-2 (Figure 5d–f) and PECAM-1 (Figure 5g–i) fluorescence intensities were not altered visually upon exposure to both positive control (TNF-α and IFN-γ) stimulation (Figure 5e,h) and polymer-coated NPs (Figure 5f,i).

### 2.4. Effects of Nanoparticles on Blood-Brain Barrier Permeability and Barrier Formation

A potential interference of NPs with the functionality of the BBB was studied, and TEER and the paracellular permeability of defined fluorescent tracers across the hBLEC monolayer were measured. The results are depicted in Figure 6. TEER was measured by means of a volt ohm meter at the end of a 24 h exposure of hBLECs to two different concentrations of either PCL- or PLLA-NPs, (2.49 × 10^−7^ μg/mL) or (24.9 µg/mL). Stimulation with 20% (*v/v*) dimethyl sulfoxide (DMSO) for 24 h was included as a positive control. No significant differences between the NP-exposed cells and non-exposed controls were detected, regardless of NP type or concentration used. DMSO treatment caused a decrease in TEER by more than 50% (Figure 6a). The paracellular permeability, as depicted by Papp (% compared to the filter; blank), was not altered after exposure to either PCL- or PLLA-NPs. Highly significant differences could only be detected when NP-exposed cells were compared to the positive control, DMSO (20%) or the blank (filter without cells) (Figure 6b,c). Papp values for 4.4 kDa dextran were in the range of 5 × 10^−6^ cm/sec to 1 × 10^−5^ cm/sec whereas permeability to LY tended to be 5 to 10 times higher.

Potential effects of PCL-NPs on the formation of a barrier in the hBLEC monolayer were investigated by means of TEER measurements at different time points and paracellular permeability to LY was measured on DIV 7 (Figure 7). Both NP types resulted in very similar effects related to uptake in the endothelial cells, the various biomarkers of inflammation and the integrity of the BBB, permeability assays were only performed for PCL-NPs. CD34+ ECs and pericytes were exposed to PCL-NPs at a concentration of (24.9 µg/mL) for 2 h prior to the establishment of the co-culture on DIV1 (Figure 7a–e, grey bars without pattern). To be able to compare to possible effects of NPs on an already formed barrier, hBLECs and pericytes were exposed in the same manner on DIV7 (Figure 7d,e, dotted filled bars). TEER measurements were carried out daily and resulted in an increase over the duration of the co-culture period (Figure 7a–d). No significant differences were detected between non-exposed controls (white bars) and co-cultures that had been exposed to NPs on DIV1 or on DIV7. All conditions differed significantly from exposure to the positive control DMSO (20%). Measurements of the paracellular permeability of LY resulted in similar findings (Figure 7e). Papp values were ranging from 1 × 10^−5^ cm/sec to 5 × 10^−5^ cm/sec.

After NP exposure to hBLECS, the NPs could not be detected in the pericytes, highlighting that they did not cross the BBB. Controls consisting of Matrigel-coated filters only confirmed that passage of NPs across the filter membrane was possible with at least 50% of NPs contained in the original stock solution being detected in the corresponding well after 24 h of exposure (Figure A3).

### 2.5. The Co-Culture Model in a 3D Setting

To get closer to the in vivo situation, the cells of the co-culture model were transferred to the cell compartments of a static 3D model. Exchange between the pericytes and CD34+ ECs with subsequent differentiation into hBLECs was possible due to migration channels connecting the cell compartments. It was shown that both cell types can handle the centrifugation step (necessary to achieve a certain concentration of cells in the medium) and attach to the Matrigel-coated PDMS and proliferate. Seeding CD34+ ECs in two to three separate steps, allowing the cells to settle in between, resulted in endothelial growth on all sides of the cell compartment and, thus the creation of an endothelial tube (Figure 8a,b). Furthermore, as in the filter membrane setting, hBLECs expressed continuous tight and adherens junctions as depicted in Figure 8c,d, with claudin 5 and VE-cadherin staining, respectively. Functional tests corroborated the formation of a tight endothelial monolayer through strongly reduced leakage of fluorescent dyes across the barrier with the time of delay correlating to the size of the tracer (Figure 8e,f). Migration of dextran into the migration channels is indicated (white line and arrows in Figure 8). After 24 h of exposure, PLLA-NPs could only be detected in cells of the bottom layer of the endothelial tube (Figure 8b) and they did not cross the BBB or move into the migration channels. NP exposure did not affect the continuity and expression of tight and adherens junction molecules or cause increased dye leakage across the BBB.

## 3. Discussion

The interactions of polymer-coated NPs used in LTS with brain ECs and the BBB were investigated in a co-culture model of hBLECs and brain pericytes in a filter membrane system and a 3D setup. Despite the prominent uptake in ECs, no other consequences of NP exposure were detected when compared to respective controls than a decrease in cell viability.

Only exposure to 24.9 µg/mL, the highest concentration of PCL- or PLLA-NPs used, caused a decrease in cell viability of both CD34+ ECs and pericytes. The effect was more marked in PCL-NP-exposed cells and it was more pronounced after exposure to NPs for 24 h than 2 h. This is in line with previous findings in rBCEC4 cells. However, these cells showed a decreased viability after exposure to lower NP concentrations as well [16]. No cytotoxic effects of PCL-NPs were detected at a concentration of 2.49 × 10^−3^ μg/mL in CD34+ ECs or pericytes. Polyisoprene NPs did not affect the viability of hCMEC/D3 cells after a 3 h exposure of up to four times the highest concentration used in this study [17], whereas Ag-NPs and TiO_2_-NPs led to a reduction in the number of viable bEnd.3 and murine astrocyte-like cells after 24 h of NP exposure. Ag-NPs exerted a stronger effect and it was shown that, although the extent of the decrease was similar in mono- and co-cultures, ECs protected the astrocyte-like cells when cultured together [19]. At concentrations similar to the ones used in this study, poly(n-butyl-cyano-acrylate) NPs exhibited no significant effects on cell viability [25]. Variations in findings are most likely due to differences in the characteristics of NPs as well as the conditions (cell models, exposure duration, NP concentrations) applied in the respective studies as it was shown that interactions of NPs and cells strongly depend on the NPs’ physicochemical properties and vary between cell types [26,27,28,29].

Both PCL- and PLLA-NPs were internalized to a high extent in CD34+ ECs and in hBLECs. Quantification by means of high-content analysis as described previously [16,30,31] could only be carried out in monocultures of CD34+ ECs as growth on permeable filter membranes and maintenance in co-culture was necessary for the differentiation of the ECs into hBLECs [24]. The membranes prevented transillumination and, thus, precluded high content analysis. After 24 h of NP exposure, about 85% of CD34+ ECs had taken up either PCL- or PLLA-NPs and the NPs were found in the cytoplasm. In the high-content analysis, a previously developed protocol was used to prevent counting NPs outside of the cells that just stick to the surface of the dish [16,30,31]. Other ECs such as hCMEC/D3 cells were reported to internalize different types of NPs like polyisoprene-based NPs [17] or superparamagnetic iron oxide NPs [32] corroborating our findings. Both studies also showed that differences in the surface coating and modifications affect the extent of NP uptake [17,32]. In hBLECs, NPs were mostly found in clusters of various sizes. This behavior as well as the extent of NP uptake, which was rather surprising for ECs, had been described previously in microglial cells [31]. In organotypic slice cultures, NPs were demonstrated to be taken up by microglial cells but not by astrocytes or neurons whereas neuronal monocultures showed internalization of PCL- and PLLA-NPs [33]. Similar to findings by Chen et al. who investigated uptake of Ag- and TiO_2_-NPs into mono- and co-cultures of bEnd.3 and astrocyte-like cells, hBLECs or pericytes had not taken up NPs after exposure to the reciprocal cell type in the co-culture setting. In general, bEnd.3 ECs were shown to have internalized a higher number of NPs than astrocytes, most of which did not overcome the BBB. This effect was more pronounced after exposure to TiO_2_-NPs compared to Ag-NPs [19]. The presence of only a single cell type in the monocultures could be the reason for NP uptake into the cells that did not internalize NPs in a co-culture setting.

As LTS is applied in ruptured cerebral aneurysms, the effects of NPs contained in the solder need to be investigated in EC models that show BBB characteristics. In agreement with Cecchelli et al., continuous expression of various AJ and TJ proteins was detected in the hBLECs [24]. An increase in TEER as well as hindered transport of the tracer molecules dextran and LY across the hBLEC monolayer further corroborated the presence of BBB properties. However, levels of TEER at around 60 Ω∙cm^2^ were lower [24] and the permeability to fluorescent tracers was higher compared to findings of others [34]. TEER values depend on the kind of device used, the handling of the cells and the temperature, factors which may explain these variations [35]. The co-culture model employed in this study displayed improved BBB characteristics compared to the previously used rBCEC4 model [16]. However, several other models of the BBB that exhibit higher TEER and lower permeability have been reported [36]. Neither the monolayer’s electrical resistance nor its permeability to the defined tracers 4.4 kDa dextran or LY or the expression of AJs and TJs were significantly altered after exposure of hBLECs or pericytes to PCL- or PLLA-NPs at different concentrations, exposure durations or time points during the co-culture period. The hCMEC/D3 model employed by Cox et al. exhibited around 2.5 x higher TEER values compared to the present co-culture model. However, similarly, none of the polyisoprene NPs investigated in their study induced any changes in TEER or permeability of the endothelial monolayer [17]. The same was shown for BMVECs in co-culture with human astrocytes after exposure to different types of NPs. Moreover, gene expression of ZO-1, JAM-2, claudin 5 and occludin remained unchanged, indicating no disturbance of the tightness of the BBB [18]. Contrary to this, in a bEnd.3 and astrocyte co-culture model, Ag- and TiO_2_-NPs led to a reduction of TEER in combination with a disruption of junctions as shown by the discontinuous expression of ZO-1 and claudin 5 [19]. Exposure of rat brain microvascular ECs to copper NPs resulted in a twofold increase in the permeability to fluorescein [34]. Contrary to the NPs used by these authors, the NPs used in this study were coated with biodegradable polymers, suggesting that inorganic material might interfere with the BBB characteristics compared to PCL and PLLA. Further supporting the toxicity of Ag-NPs, perivascular edema and swollen astrocytic end-feet were found in adult rats. Although no ultrastructural anomalies could be detected in pericytes and TJ complexes, changes in the expression of the junctional proteins claudin 5, occludin and ZO-1 were seen. A concentration-dependent increase in permeability to sucrose was seen after a 2 h exposure of hBLECs to PLGA-NPs [20]. These NPs were about twice the size of the NPs used in this study. Besides the difference in size, the NPs were designed to bind to transferrin and therefore, different physicochemical properties might explain the difference in permeability.

Pro-inflammatory stimuli were demonstrated to lead to induction and increase of expression of various endothelial adhesion molecules and markers [11] which are regulated by the NF-κB signaling pathway [37]. Exposure to PCL- and PLLA-NPs, however, did not cause activation of NF-κB or stimulate expression or upregulation of several adhesion molecules and markers which is in agreement with previously reported findings [16,38]. However, 2-h incubations with nanoparticles were investigated and it cannot be ruled out that a longer incubation time, for example, 16 h would have increased the inflammation markers investigated. Au-NPs were shown to cause a time-dependent increase in various pro-inflammatory markers such as soluble ICAM and VCAM [39]. Upregulated expression of these markers was also demonstrated after exposure to alumina NPs. The effect on VCAM was found not to be caused by activation of NF-κB, suggesting that other pathways are also involved in the modulation of these adhesion molecules [40].

It has been shown that 3D models have various advantages to study either uptake mechanisms and/or permeability changes of the BBB after nanomaterial exposure [41,42]. It was shown that both CD34+ ECs and pericytes could be centrifuged and seeded into the cell compartments of the 3D models. Maintaining the cells in culture for 7 days allowed for differentiation of the CD34+ ECs into hBLECs, expressing TJ markers and hindering dye leakage, similar to the established co-culture model in the filter membrane system [24]. Hence, these two cell types can be used as a model of the BBB in a 3D setting. Findings in the 3D model further corroborate the lack of effects of NPs on the integrity of the BBB, that is, no reduction of expression of tight and adherens junctions or increased dye leakage across the endothelial barrier. The polymer-coated NPs used in this study were not found to migrate across the BBB. Contrary to these findings, Li et al. showed that indoor nanoscale particulate matter was able to translocate across the BBB in their 3D microfluidic chip. Furthermore, exposure to these particles led to induction of inflammation through reactive oxygen species and pathways involved in oxidative stress [43]. Differences are most likely due to variations in physicochemical characteristics between the investigated NPs. Previously, it has already been demonstrated that the polymer-coated NPs used in this study do not cause oxidative stress in ECs [44].

Results of this study corroborate previous findings that had been made in a more simplistic model of the BBB and further support the safe use of polymer-coated NPs in LTS. Apart from an effect on cell viability at concentrations much higher than those expected to be found after slow degradation of the scaffold and NP release into the brain tissue, no adverse effects on morphology, function and integrity of the BBB were detected. Potential induction of inflammation and upregulation of endothelial adhesion molecules need to be investigated in more detail but preliminary data showed no effect of PCL- or PLLA-NPs on hBLECs. The integration of another cell type such as neurons as well as the introduction of physiological flow would move the 3D model even closer to the in vivo situation.

## 4. Materials and Methods

### 4.1. Cell Culture of the Human Brain-Like Endothelial Cells

Two cell types, brain pericytes and human brain-like endothelial cells (hBLECs), were used in a co-culture model in this study. This model was established and described in detail by Pedroso et al. and Cecchelli et al. [24,45]. Briefly, human CD34+ cells were isolated from human umbilical cord blood with parental consent and in compliance with the French legislation. The French Ministry of Higher Education and Research approved the protocol used for subsequent differentiation of these cells into CD34+-derived endothelial cells (CD34+ ECs) (CODE-COH Number DC2011-1321). To induce BBB characteristics in CD34+ ECs, a co-culture system with bovine pericytes was set up. Briefly, cells were expanded in 100 mm petri dishes (Corning, VWR, Switzerland) coated with 0.2% (*w/v*) pig skin gelatin type A (Sigma, Switzerland) in endothelial cell medium (ECM, ScienCell, Chemie Brunschwig, Switzerland) supplemented with 5% fetal bovine serum (FBS, Gibco, France), 1% Endothelial Cell Growth Supplement (ECGS, ScienCell, Chemie Brunschwig, Switzerland) and penicillin, (50 units/mL) and streptomycin, (50 μg/mL) (ScienCell, Chemie Brunschwig, Switzerland) (the medium was then referred to as ECM-5) at 37 °C and 5% CO_2_. The cells reached full confluence after 2 days in culture. Pericytes were then seeded into gelatin-coated multiwell plates (12-/24-well plates from Corning, 96-well plates from Greiner Bio-One, all from VWR, Switzerland) and CD34+ ECs were transferred onto filter membranes (Transwell permeable supports with 0.4 μm polycarbonate membrane, 12- or 24-well format, Corning, VWR, Switzerland) coated with Matrigel (Corning, VWR, Switzerland). The co-cultures were started by transferring the filters with the ECs to the wells containing the adhered pericytes. The cells were kept in co-culture for 7 days to achieve induction and expression of BBB characteristics and, thus transformation of CD34+ ECs into hBLECs. The medium was changed every other day.

### 4.2. Expression of Blood-Brain Barrier Characteristics in hBLECs

Immunofluorescence microscopy (IF) (Zeiss Axio Imager Z1 coupled with an Apotome 1 (Carl Zeiss Vision Swiss AG, Feldbach, Switzerland)) and the Philips CM12 transmission electron microscopy (TEM) (FEI, Eindhoven, The Netherlands) were used to assess the expression of BBB characteristics in the hBLECs on DIV7 of the co-culture period (Figure A4). The tight junction (TJ) proteins claudin 5 (Figure A4a) and junctional adhesion molecule A (JAM-A) (Figure A4b) were shown to be expressed continuously in cell-cell contacts. Furthermore, the adherens junction molecule vascular endothelial cadherin (VE-cadherin) (Figure A4c) as well as the TJ-associated scaffolding protein ZO-1 (Figure A4d) were also detected and continuously expressed. Formation of these cell junctions was further corroborated by TEM (Figure A4e). To functionally demonstrate the induction of barrier properties in the hBLEC monolayer, TEER was measured daily during the 7 days of co-culture. A significant daily increase was found reaching a plateau on DIV6 and DIV7 of the co-culture period, resulting in TEER values of around 60 Ω∙cm^2^ (Figure A4f).

### 4.3. Nanoparticles

Two types of silica-based polymer-coated NPs, namely silica-ICG/poly(ε-caprolactone) (PCL) and silica-ICG/poly(ε-caprolactone-poly(L-lactide)) (PLLA) NPs, were manufactured and provided by Dr. Uwe Pieles, Department of Chemistry and Bioanalytics, Academy of Life Science, Switzerland. A silica-core was encapsulated by a layer of PCL/ICG, followed by a polymer surface coating with either PCL or PLLA. For simplified visualization by means of fluorescence microscopy, NPs with a rhodamine-doped core were constructed as well. Non-fluorescent and fluorescent NPs exhibited the same surface properties. The physicochemical properties of both nanoparticle types are given in Table 1 [46]. Stock solutions of both NP types were prepared by dissolving the dried NP powder in DMSO (0.25%; Sigma, Switzerland) and Dulbecco’s Phosphate-Buffered Saline (DPBS) (Life Technologies, UK). Prior to incubation with cells, the stock solutions were sonicated to achieve homogenous distribution of the NPs in solution. Three cycles of sonication (4 min at 30% amplitude on ice followed by a 5 min pause) were carried out. For the experiments, PCL- and PLLA-NP-stock solutions were diluted 1:10 in ECM-5 or cell culture medium with only 1% FBS (ECM-1), resulting in NP numbers of 2.9 × 10^10^ PCL-NPs in 1 mL culture medium and 2.6 × 10^10^ PLLA-NPs in 1 mL culture medium.

For PCL-NPs, a stock solution of (2.6 × 10^11^ PCL-NPs/mL) was made, using 0.0025% DMSO (Sigma, USA) in Dulbecco’s phosphate buffered saline (DPBS, Gibco, Life Technologies, UK). The stock solution was sonicated three times for five min with cooling steps in between to enable homogeneous NP suspension just before treatment of the cells. The final concentration of PCL-NPs (2.6 × 10^10^ PCL-NPs/mL), corresponding to (24.9 µg/mL), used for all experiments was obtained by dilution of the stock with culture medium containing 1% FBS as previously reported [33]. Concentrations of 2.9 × 10^6^ PCL-NPs/2.6 × 10^6^ PLLA-NPs and 2.9 × 10^2^ PCL-NPs/2.6 × 10^2^ PLLA-NPs in 1 mL cell culture medium were used in this study. Overall, these two different NP numbers correspond to concentrations of 2.49 × 10^−3^ μg/mL and 2.49 × 10^−7^ μg/mL, respectively. These NPs were designed to be used in LTS. Hence, the scaffold that contains these nanoparticles will degrade slowly over time. This study is part of a large project that has been going on for several years. To keep the conditions as consistent as possible, time points of NP exposure and concentrations of NPs were used in accordance with what had been done in previous studies [31,33].

### 4.4. Nanoparticle Uptake

The uptake of PCL- and PLLA-NPs into hBLECs was assessed by means of TEM and structured illumination microscopy and quantified using high-content analysis with the INCell Analyzer 2000 (GE Healthcare Life Sciences, USA). Cells were cultured in co-cultures as described above under ‘Cell culture of the human brain-like endothelial cells’ (12-well format). On day in vitro (DIV) 7, hBLEC cells were fixed with pre-warmed 2.5% glutaraldehyde in 0.1M cacodylate buffer, pH 7.4 (both Merck, Switzerland) for TEM or 4% paraformaldehyde (PFA; Sigma, Switzerland) for detection via immunofluorescence (IF) after exposure to PCL- or PLLA-NPs at a concentration of 24.9 µg/mL for 24 h. TEM micrographs were acquired as described previously [47]. Additional exposures of hBLECs to NPs for 2 h or 30 min were carried out for structured illumination microscopy with a Zeiss AxioImager Z1 plus Apotome 1.

To be able to quantify the uptake of NPs, CD34+ ECs or pericytes were seeded at 25,000 cells or 10,000 cells in gelatin-coated wells (96-well plate) on DIV0, respectively. Exposure to PCL- or PLLA-NPs at a concentration of 24.9 µg/mL were started on DIV3 (24 h exposure) or on DIV4 (2 h or 30 min exposure) at 37 °C and 5% CO_2_ and stopped together on DIV4. Subsequently, quantitative evaluations were completed by high-content analysis with the INCell Analyzer 2000 [30,31]. The INCell Analyzer allows high-content analysis. We investigated 30 pictures/well × 3 replicated × 3 experiments. The analysis of one experiment/time/nanoparticle type/ per setting includes the analysis of about 50,000 cells.

### 4.5. Cell Viability and Cytotoxicity Assays

A possible effect of these two types of NPs on the viability of CD34+ ECs and pericytes was assessed by means of the methylthiazolyldiphenyl-tetrazolium bromide (MTT) (Sigma, Switzerland) assay. Cell viability was investigated at full confluence of both CD34+ ECs and pericytes. CD34+ ECs were seeded at 25,000 cells and pericytes at 10,000 cells in separate wells in a gelatin-coated 96-well plate on DIV0. On DIV3, exposure of both cell types to three different concentrations of PCL- and PLLA-NPs for 24 h was started. A 2 h NP exposure was carried out on DIV4. Controls (non-exposed CD34+ ECs and pericytes) were done in parallel. All cells were incubated with MTT dissolved in phosphate-buffered saline (PBS) (Life Technologies, UK) at a final concentration of [0.5 mg/mL] for 4 h at 37 °C and 5% CO_2_ at the end of the NP exposure. After removal of the cell culture medium, the remaining MTT-formazan was dissolved in DMSO. Absorbance was measured at 540 nm using a plate reader (Synergy H1, BioTek, Switzerland).

Staining with propidium iodide (PI, (1 mg/mL); Sigma, Switzerland) in combination with Hoechst (Invitrogen, Switzerland) was performed to assess the cytotoxic effect of non-fluorescent PCL-NPs. This assay was combined with a second cell viability assay. Hereby, the immediate effects (DIV1) of non-fluorescent PCL-NPs on CD34+ ECs and pericytes as well as potential consequences several days after NP exposure (DIV4) were examined. On DIV0, 30’000 CD34+ ECs and 10,000 pericytes were seeded in separate wells in gelatin-coated 96-well plates. A 24 h exposure to PCL-NPs was started about 3 h afterwards, allowing the cells to adhere to the plates. Stimulation with staurosporine (diluted in DMSO to a final concentration of 2 µM; Sigma, Switzerland) for 24 h was used as a positive control on DIV1 and on DIV4. Non-exposed or -stimulated cells were used as controls. The NP exposure was stopped in all plates on DIV1 by exchanging the medium. Cytotoxicity of NPs was compared relatively to staurosporine. To assess an immediate effect of PCL-NPs on the cells, the MTT and the PI assay were carried out immediately in half of the wells. The remaining half was cultured until DIV4 on which cell viability and cytotoxicity was measured in these cells as well. The MTT assay was performed as described above. PI was diluted in 1X binding buffer (Sigma, Switzerland) to a final concentration of 20 µg/mL. The nuclear dye Hoechst was added as a counterstain (final concentration: 1 µg/mL). Cells were washed twice in DPBS, followed by a 10 min incubation period with the staining solution at room temperature (RT) in the dark before the fluorescence signal of Hoechst (excitation: 360 nm, emission: 440 nm) and PI (excitation: 530 nm, emission: 620 nm) was detected by a plate reader (Synergy H1, BioTek, Switzerland). Afterwards, live cell imaging of the cells and high-content analysis with subsequent quantification with the INCell Analyzer 2000 were carried out. High-content analysis refers to the combination of automated microscopy and automated, quantitative analysis of specifically labelled micrographs. It provides objective, accurate and statistically significant data on complex cellular features.

### 4.6. Immunofluorescence Microscopy

As described above, CD34+ ECs were grown in co-culture for 7 days until differentiation into hBLECs (12- and 24-well format as well as 3D models). Assessments of NP uptake (described above) and adherens (AJ) and tight junction (TJ) formation as well as of effects of NP exposure on the expression of NF-κB and on junctional proteins as part of the permeability assays (described below) were carried out in co-cultures after fixation with ice-cold methanol for 10 s (claudin antibodies) or cold 4% PFA (all other antibodies) at RT, followed by two washing steps with DPBS. The filter membranes were either left in the supporting Transwell systems or cut out with a scalpel and transferred to a plate (96- or 24-well plate) for staining. The staining procedure remained the same. Staining solution was injected into the medium reservoirs in the static 3D models. After blocking with 10% horse serum in 0.4% Triton-PBS for 2 h at RT, the samples were incubated with a respective primary antibody overnight at 4 °C (Table 2). Subsequently, samples were washed with PBS (four times, 10 min each) and incubated with the corresponding secondary antibodies as well as Acti-stain phalloidin (Table 2) for 2 h at RT. Following another washing step with PBS for four times, 10 min each, the filter membranes containing the hBLEC monolayer were either cut from the Transwell system or removed from the plate. With the cells facing up, they were then mounted on glass slides using Glycergel Mounting Medium (Dako, Denmark/USA). To assess possible induction of inflammation by NPs, the expression of NF-κB and its phosphorylated form was evaluated after exposure to fluorescent PCL-NPs at a concentration of 24.9 µg/mL for 24 h, started on DIV6 and stopped on DIV7. Lipopolysaccharide (LPS) was used as a positive control at a concentration of 1 µg/mL for 2 h on DIV7. For investigation of the interactions of NPs with several adhesion molecules, hBLECs were exposed to fluorescent PLLA-NPs at a concentration of 24.9 µg/mL for 2 h on DIV7. Stimulation with a combination of TNF-α and IFN-γ (final concentration: 1 ng/mL) for 16 h was used as positive control. Both substances were added to the wells on DIV6 and removed on DIV7 alongside with the NP-containing medium in NP-exposed wells. Staining for PECAM-1, E-selectin and P-selectin was performed after fixation of the hBLECs with cold 1% PFA for 10 min at RT and subsequent washing. Filter membranes were cut and transferred to wells of a 96-well plate in which they were incubated with antibodies as outlined above before being mounted on glass slides. ICAM-1, ICAM-2 and VCAM-1 stainings were performed in live cells similarly as previously described [48]. Briefly, cell culture medium was substituted with the respective antibody diluted in ECM-5 and incubated for 15 min at 37 °C and 5% CO_2_. The filters were then washed twice with warm DPBS before fixation with cold 1% PFA. Following two washing steps with PBS, the cells were permeabilized and blocked for 10 min at RT and then incubated with the corresponding secondary antibody and Acti-stain 670 phalloidin (Table 2) for 60 min at RT. Finally, after another washing step with DPBS, the filter membranes were mounted as illustrated above.

The AxioImager was used to visualize immunostaining of fixed hBLECs grown in co-culture that is, on membranes and to demonstrate qualitatively the uptake of PCL- and PLLA-NP but not to quantify NP uptake. Quantification with the INCA could only be performed on monocultures of CD34+ ECs that were grown without membranes.

Barrier formation and impact of NP exposure on the BBB were assessed by live cell imaging with the confocal microscope (Nikon Eclipse Ti-E, Nikon, Switzerland or Olympus FV 3000, Olympus, Switzerland).

### 4.7. Transendothelial Electrical Resistance and Permeability Studies

To evaluate possible effects of NPs on an established BBB, cells were cultured as described above (12-well format) and were exposed to fluorescent PCL- or PLLA-NPs at concentrations of 2.49 × 10-7 μg/mL or 24.9 µg/mL for 24 h on DIV7. Cells stimulated with 20% (*v/v*) DMSO (Sigma, Switzerland) for 24 h to induce loss of barrier integrity were used as positive control. To monitor the formation of a cellular barrier, as well as the integrity of the established hBLEC monolayer on DIV7, the transendothelial electrical resistance (TEER) was measured daily over the duration of the co-culture period as well as at the end of the exposure to NPs or the stimulation with the positive control using the Millicell-ERS volt ohm meter (Merck Millipore, Switzerland). Equation (1) was used to calculate the TEER (Ω∙cm^2^) of the hBLEC monolayer [49]:(1)TEER=(RTotal−RBlank)×AMembrane
RTotal denotes the resistance across the hBLEC monolayer on the Matrigel-coated filter membrane (Ω), RBlank represents the resistance across an empty, Matrigel-coated filter membrane (no cells) (Ω) and AMembrane is the filter membrane’s surface area (cm^2^).

Subsequently, a permeability assay was performed at the end of NP exposure on DIV7 as described previously [24]. Transport buffer (TB) was obtained by mixing Hank’s Balanced Salt Solution (HBSS; Sigma, Switzerland) with HEPES buffer (final concentration: 10 mM; Life Technologies, Switzerland). The fluorescent tracers, Lucifer yellow (LY) and 4.4 kDa tetramethylrhodamine isothiocyanate (TRITC) dextran (both Sigma, Switzerland) were diluted in TB to final concentrations of 50 µM and 0.5 mg/mL, respectively. The cell culture medium was removed from the filters and was substituted with the tracer solution after a transfer to 12-well plates containing TB. The filters were incubated on a microplate shaker for 20 min at RT before being transferred to new wells (filled with fresh TB). This step was repeated three times, resulting in a total incubation time of 60 min. Samples from all donor (filters) and receiver (wells at three different time points) solutions were taken and fluorescence was measured at 425 nm (excitation), 530 nm (emission) and 550 nm, 580 nm for LY and 4.4 kDa TRITC dextran, respectively, with a plate reader (Synergy H1, BioTek, Switzerland).

To describe potential effects of NP exposure on the formation of a tight barrier in the hBLEC monolayer, a second set of experiments was carried out as follows: pericytes and CD34+ ECs were seeded into gelatin-coated wells and Matrigel-coated filters (12-well format), respectively. Cells were kept in ECM-5 in monocultures for 1 day, followed by a 2 h exposure to non-fluorescent PCL-NPs at a concentration of [24.9 µg/mL] in CD34+ ECs or pericytes. Subsequently, co-cultures were set up in a manner such that either CD34+ ECs or pericytes or both cell types had been exposed to NPs. For direct comparison, another set of co-cultures was exposed to non-fluorescent PCL-NPs at a concentration of 24.9 µg/mL for 2 h on DIV7 of the co-culture period in a similar fashion (hBLECs or pericytes only, both cell types). On the same day, a 1 h exposure to 20 % (*v/v*) DMSO was included as a positive control as it disrupts the junctions between cells. In this set of experiments, PCL-NPs were diluted in cell culture medium containing only 1% FBS instead of 5% FBS. Both cell types were maintained in ECM-5 at all times apart from during NP exposure. TEER measurements were conducted and values were calculated (Equation (1)) as described above. The permeability assay was carried out as previously described with only LY at a concentration of 50 µM. Fluorescence was detected at 425 nm (excitation) and 545 nm (emission) using a plate reader (Synergy H1, BioTek, Switzerland).

Subsequently, calculations of the apparent permeability coefficient (Papp; cm/s) were performed according to Equation (2):(2)$$\mathrm{Papp}=\frac{\left(k\times \mathrm{VB}\right)}{\left(A\times 60\right)}$$
k signifies the rate of transport defined as the slope obtained after applying linear regression to the cumulative fraction absorbed (FAcum) plotted versus time (s-1), VR denotes the volume in the receiver chamber (wells, mL) and A is the surface area of the filter membrane (cm^2^). Equation (3) was used to calculate FAcum:(3)$$\mathrm{FAcum}=\frac{\mathrm{CRi}}{\left(\mathrm{CDi}\right)}$$
CRi denotes the concentration in the receiver chamber (well, µM) at the end of interval i, CDi represents the concentration in the donor chamber (filter, µM) at the beginning of interval i.

### 4.8. 3-Dimensional Blood-Brain Barrier Model

The 3D model was established and thoroughly described by Cho et al. and the molds used in this study were created and kindly provided by Dr. Hansang Cho (Mechanical Engineering and Engineering Science, University of North Carolina at Charlotte, North Carolina, USA) [41]. The model consists of a single layered microfluidic platform made from polydimethyl-siloxane (PDMS) that contains two cell compartments which are connected by smaller so-called migration channels. Briefly, silicone elastomer and curing agent were mixed at a ratio of 10:1 (Sylgard 184 silicone elastomer kit, Dow Corning, Germany). After removal of air bubbles contained in the mixture by applying a vacuum, the mixture was poured on a microstructured mold and kept in an oven overnight at 60 °C for curing of the PDMS. Subsequently, single models were cut out from the cured PDMS, dust was removed and models were bonded to µ slides with a glass coverslip bottom (thickness 0.17 mm; µ-Slide 2 Well Glass Bottom, ibidi, Vitaris, Switzerland) by means of plasma bonding with a PE50XL Plasma Cleaner (Inseto (UK) Limited, UK). Directly afterwards, cell compartments and migration channels were coated with Matrigel to allow for cell adhesion. Entrance of the Matrigel into all structures of the models was achieved by shortly applying a vacuum. Finally, CD34+ ECs and pericytes were seeded into separate cell compartments at high densities (about 30 million and 20 million cells, respectively) using gel loading pipet tips (VWR, Switzerland) and confined to the respective cell compartment by surface tension. CD34+ ECs were seeded in three steps to have cells adhere to the upper part (2 cell seedings) of the cell compartment as well as to the lower, bottom part, enabling vessel formation as cells can grow down and up the sides. The cells were gently added to the medium reservoir of the respective cell compartment and given some time to settle. Excess cell suspension was removed from the medium reservoir, leaving cells inside the compartment. The models were then incubated upside down at 37 °C and 5% CO_2_ for 30 min before repeating this step. Finally, a third cell seeding step was followed by incubation of the models in the correct position. Cell compartments were 50 µm in height, 250 µm in width, migration channels were 5 µm in height, 10 µm in width and 600 µm in length. The models were maintained at 37 °C and 5% CO_2_ for 7 days with cell culture medium (ECM-5) changes every day.

For functional assessment of the barrier formation and the impact of NP exposure on the BBB, the leakage of three differently sized fluorescent tracers, namely fluorescein isothiocyanate (FITC) dextran (4 kDa, 40 kDa and 70 kDa; all Sigma, Switzerland) across the EC monolayer was assessed by means of live cell imaging with a confocal microscope (Olympus FV 3000, Olympus, Switzerland). Medium in the EC compartment was substituted with the tracer solution (final concentrations in TB: 4 kDa dextran (50 µM), 40 kDa dextran (10 µM), 70 kDa dextran (6 µM)) and the model was immediately transferred to the microscope to record the leakage of the respective tracer from the EC compartment into the migration channels over time by capturing an image every 30 s for 20 min.

### 4.9. Statistical Analysis

Statistical analysis was performed using GraphPad Prism (GraphPad Software Inc., La Jolla, USA). All experiments were assessed by one-way ANOVA, followed by Tukey’s multiple comparison test. *p*-values ≤ 0.05 were considered statistically significant. Each experiment was carried out in triplicates and repeated at least two to three times. Results depicted in the figures are given as mean and error bars specify standard error of the mean (SEM).

## Figures and Tables

**Figure 1 ijms-22-01657-f001:**
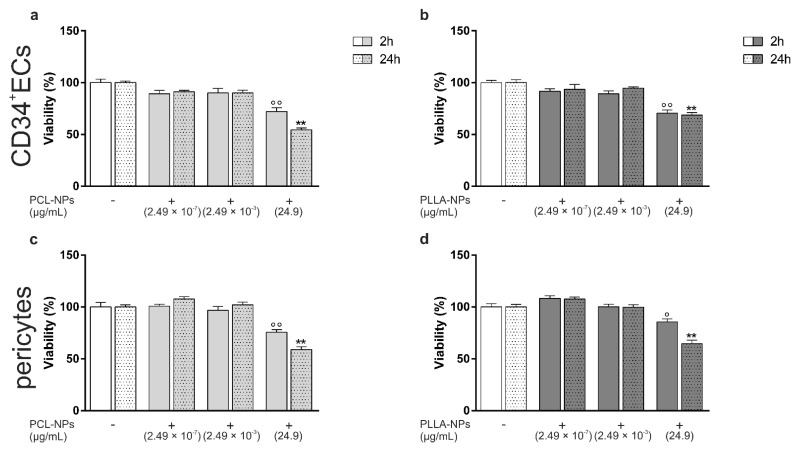
Concentration- and time-dependent effects of silica-ICG/poly(ε-caprolactone) (PCL)- (**a**,**c**) and silica-ICG/poly(ε-caprolactone-poly(L-lactide)) (PLLA)- (**b**,**d**) Nanoparticles (NPs) on the viability of CD34+ endothelial cells (ECs) (**a**,**b**) and pericytes (**c**,**d**). Both cell types were exposed to NPs for 2 (filled bars) and 24 h (dotted bars) at three different concentrations (*n* = 3). Exposure to NPs is marked with a +, cells not exposed to NPs with a −. Error bars represent SEM. Significant differences between NP-exposed and non-exposed controls (white bars) are labeled with circles (°) for 2 h of NP exposure and with asterisks (*) for 24 h of NP exposure (° = *p* ≤ 0.05; °°/** = *p* ≤ 0.0001).

**Figure 2 ijms-22-01657-f002:**
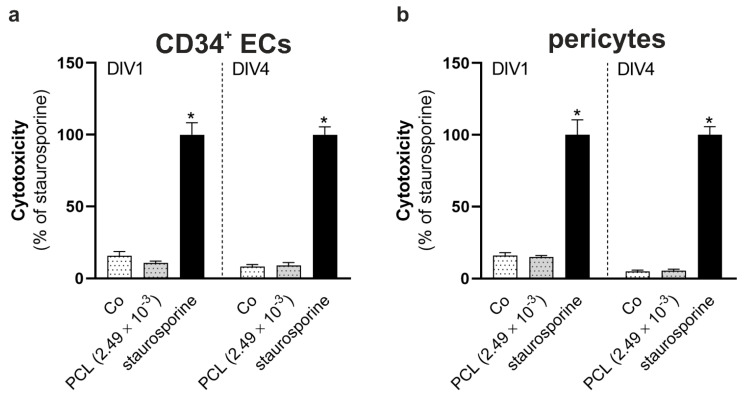
Cytotoxicity of PCL-NPs after exposure for 24 h on day in vitro (= DIV) 1. CD34+ ECs (**a**) and pericytes (**b**) were exposed to PCL-NPs at a concentration of 2.49 × 10^−3^ μg/mL for 24 h on DIV1 (dotted bars) (*n* = 3). Stimulation with staurosporine in dimethyl sulfoxide (DMSO) (final concentration: 2 µM) for 24 h was used as positive control (black bars) and carried out on DIV1 or on DIV4 before detecting cytotoxicity. Non-exposed or –stimulated cells were used as control (Co). Error bars represent SEM. Significant differences compared to the positive control are labeled with asterisks (*) (* = *p* ≤ 0.0001).

**Figure 3 ijms-22-01657-f003:**
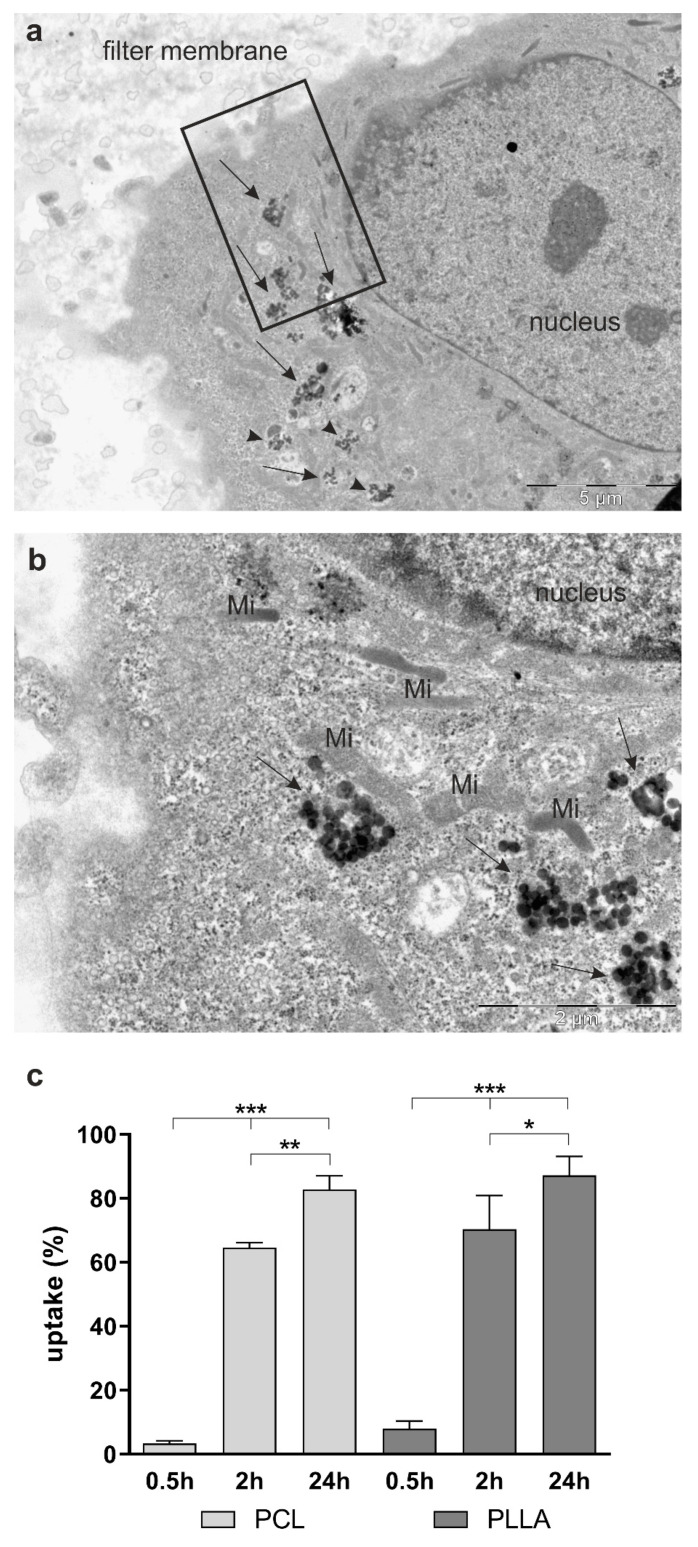
Uptake of PCL- and PLLA-NPs in CD34+ ECs and human brain-like endothelial cells (hBLECs). Representative transmission electron microscopy (TEM) micrographs of hBLECs demonstrate the uptake of PCL-NPs into the cells after exposure for 24 h (**a**,**b**). NPs were seen freely in the cytoplasm (arrows) or in vesicles, surrounded by a membrane (arrow heads). (**b**) (scale bar: 2 µm) represents a higher magnification of the marked part (black frame) in (**a**) (scale bar: 5 µm). Mi = mitochondrion. The uptake of both types of NPs into CD34^+^ ECs was quantified by means of high-content analysis and an uptake of 100% would indicate that all cells internalized NPs (**c**) (*n* = 3). The cells were exposed to PCL- or PLLA-NPs for 0.5, 2 or 24 h (= 0.5 h, 2 h or 24 h). Error bars represent SEM. Significant differences are labeled with asterisks (*) (* = *p* ≤ 0.01, ** = *p* ≤ 0.001, *** = *p* ≤ 0.0001). Concentrations of PCL- or PLLA-NPs were (24.9 µg/mL) in (**a**–**c**).

**Figure 4 ijms-22-01657-f004:**
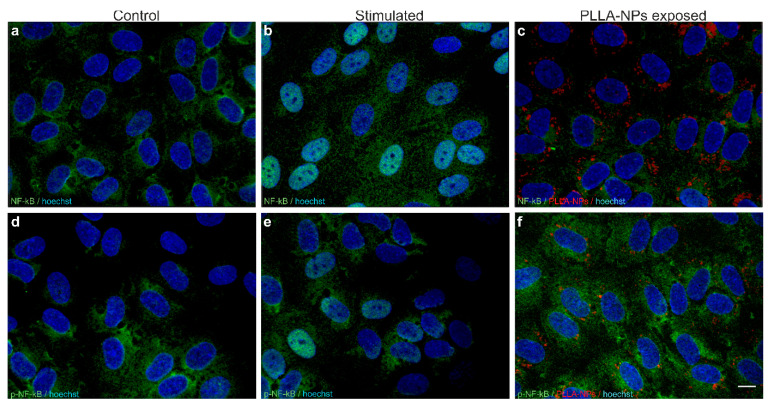
Representative microscopic images of effects of polymer-coated NPs on the expression of NF-κB (**a**–**c**) and p-NF-κB (**d**–**f**) (green). Non-exposed cells were used as control (**a**,**d**). Stimulation with lipopolysaccharide (LPS) (1 µg/mL) for 2 h on DIV7 (**b**) or a combination of tumor necrosis factor (TNF)-α (1 ng/mL) and IFN-γ (1 ng/mL) for 16 h, starting on DIV6 (**e**) was carried out as positive controls. hBLECs were exposed to PLLA-NPs for 2 h on DIV7 at a concentration of (24.9 µg/mL) (**c**,**f**). Cell nuclei were counterstained with Hoechst (blue). Scale bars: 10 µm in each plane.

**Figure 5 ijms-22-01657-f005:**
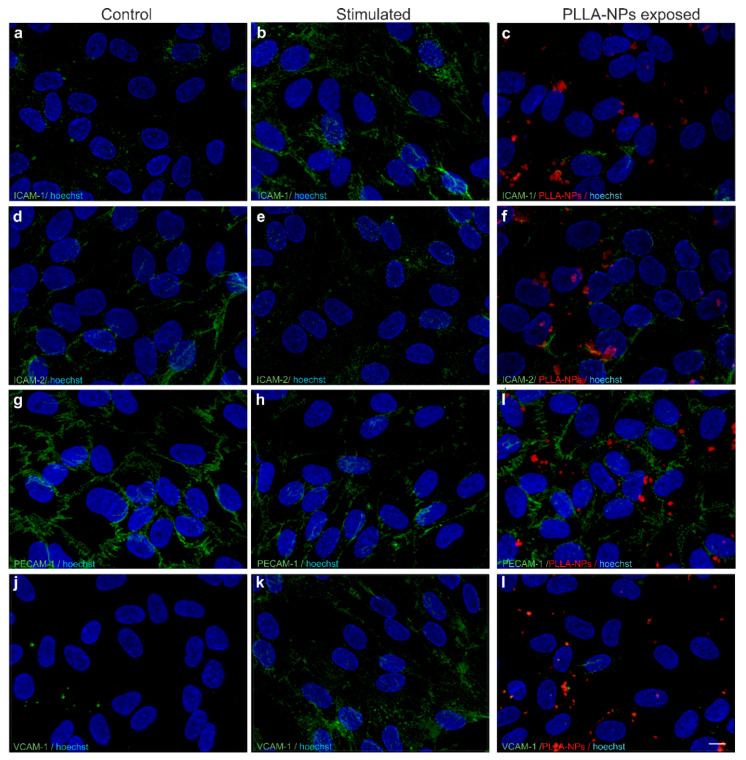
Representative microscopic images of effects of polymer-coated NPs on the expression of ICAM-1(**a**–**c**), ICAM-2 (**d**–**f**), PECAM-1 (**g**–**i**) and VCAM-1 (**j**–**l**) (green). Non-exposed cells were used as control (**a**,**d**,**g**,**j**). Stimulation with a combination of TNF-α (1 ng/mL) and IFN-γ (1 ng/mL) for 16 h, starting on DIV6 (**b**,**e**,**h**,**k**) was carried out as positive controls. hBLECs were exposed to PLLA-NPs for 2 h on DIV7 at a concentration of (24.9 µg/mL) (**c,f**,**i**,**l**). Cell nuclei were counterstained with Hoechst (blue). Scale bars: 10 µm in each plane.

**Figure 6 ijms-22-01657-f006:**
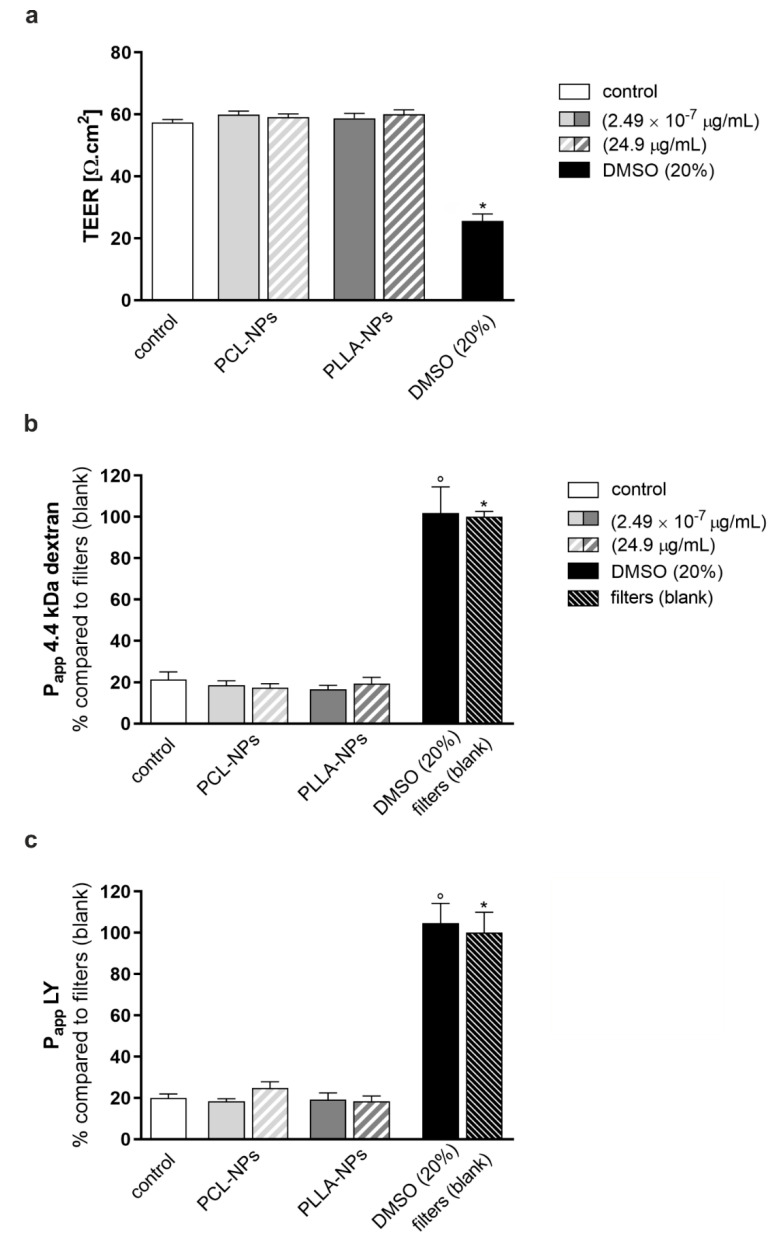
Effects of PCL- and PLLA-NPs on the permeability of the blood-brain barrier. hBLECs were exposed to the two types of NPs for 24 h on DIV7 of co-culture. 20% (*v/v*) DMSO was used as a positive control. Transendothelial electrical resistance (TEER) was measured with a volt ohm meter (**a**). Fluorescent 4.4 kDa dextran (**b**) and Lucifer yellow (LY) (**c**) were used to calculate the apparent permeability coefficient (P_app_) across the hBLEC monolayer (*n* = 3). Control (white bars) = non-exposed cells. Error bars represent SEM. Statistically significant differences compared to DMSO 20% (black bars) or the blank (filters without cells) (stripped bars) are labeled with circles (°) or asterisks (*), respectively (°/* = *p* ≤ 0.0001).

**Figure 7 ijms-22-01657-f007:**
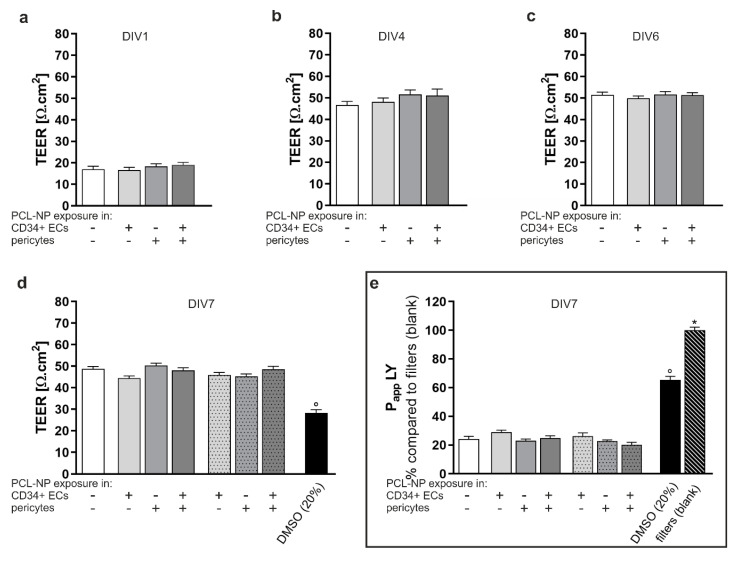
Effects of PCL-NPs on barrier formation of the hBLEC monolayer. Analyzes were performed by measurements of transendothelial electrical resistance (TEER) and the permeability to Lucifer yellow (LY); given as the apparent permeability coefficient P_app_). CD34^+^ ECs TEER was measured every day of the co-culture; results are depicted for DIV1 (**a**), DIV4 (**b**), DIV6 (**c**) and DIV7 (**d**). CD34^+^ ECs or pericytes were exposed (+) to PCL-NPs for 2 h on day in vitro (= DIV) 1 (uniform filled bars), before establishment of the co-culture. hBLECs and pericytes were exposed (+) to PCL-NPs for 2 h on DIV7 (dotted bars). Non-exposed cells (-) were used as a control (Co) (white bars) (*n* = 3). Stimulation with 20% (*v/v*) DMSO for 1 h on DIV7 was included as positive control (black bars). The permeability to LY was assessed on DIV7 (**e**). Concentration of PCL-NPs was (24.9 µg/mL). Error bars represent SEM. Significant differences compared to the positive control or blank (striped filled bar) are labeled with circles (°) or asterisks (*), respectively (°/* = *p* ≤ 0.0001).

**Figure 8 ijms-22-01657-f008:**
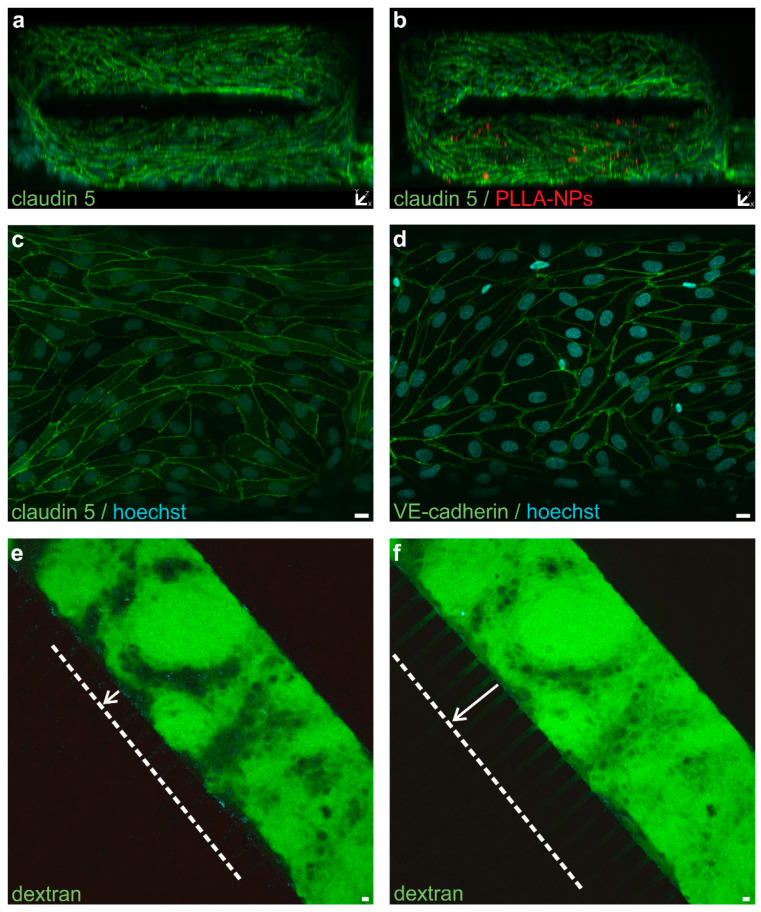
Representative confocal microscopic images of the 3D co-culture model of the blood-brain barrier. Only the endothelial cell compartment is depicted. Confluent monolayers of hBLECs line the lumen of the 3D tube structure in the 3D reconstruction (**a**,**b**). Cells were stained for claudin-5 (green) (**a**,**b**). hBLECs were exposed to PLLA-NPs (24.9 µg/mL) (red) for 24 h (**b**). Formation of claudin 5 (**c**) and VE-cadherin (**d**) (green) after a co-culture period of 7 days. Leakage of 40 kDa dextran (10 µM) (green) across the endothelial layer into the migration channels after addition of the tracer solution (**e**) and 12 min later (**f**). The dotted white lines and the white arrows mark the dextran migration into the migration channels. Cell nuclei were counterstained with Hoechst (blue). Scale bars: 10 µm.

**Table 1 ijms-22-01657-t001:** Physicochemical characteristics nanoparticles (NPs).

Characteristics	Silica-ICG/poly(ε-caprolactone) (PCL) NPs	Silica-ICG/poly(ε-caprolactone-poly(l-lactide)) (PLLA) NPs
Silica-core encapsulated by	PCL/ICG	PCL/ICG
polymer surface coating	PCL	PLLA
Rhodamine-doped core	yes	yes
Size	90 nm	95 nm
Zeta potential	−25.4 mV	−15.9 mV

**Table 2 ijms-22-01657-t002:** Antibodies.

	Antibody	Host	Company	Method	Dilution
primary	claudin 3	rabbit	Abcam	IF	1:50
	claudin 5	mouse	Santa Cruz	IF	1:100
	claudin 5	rabbit	Abcam	IF	1:100
	JAM-A	mouse	Santa Cruz	IF	1:100
	ZO-1/TJP1	mouse	ThermoFisher	IF	1:100
	VE-cadherin	mouse	Santa Cruz	IF	1:100
	NF-κB	rabbit	Cell Signaling	IF	1:100
	phospho-NF-κB	rabbit	Cell Signaling	IF	1:100
	ICAM-1	mouse	BioLegend	IF	1:100
	ICAM-2	mouse	Fitzgerald	IF	1:100
	VCAM-1	mouse	BD Pharmingen	IF	1:100
	PECAM-1	mouse	Zymed	IF	1:100
secondary	anti-rabbit IgG AF 488	goat/donkey	Invitrogen	IF	1:250
	anti-mouse IgG AF 488	donkey	Invitrogen	IF	1:250–1:500
	anti-mouse IgG AF 647	donkey	Invitrogen	IF	1:250
other	Hoechst	-	Life Technologies	IF	1:10,000
	Acti-stain 488 phalloidin	-	Cytoskeleton	IF	1:50
	Acti-stain 555 phalloidin	-	Cytoskeleton	IF	1:50
	Acti-stain 670 phalloidin	-	Cytoskeleton	IF	1:50

AF = Alexa Fluor; IF = immunofluorescence.

## Data Availability

There are no additional data supporting reported results.

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
