# Peer review of "Time-Dependent Internalization of Polymer-Coated Silica Nanoparticles in Brain Endothelial Cells and Morphological and Functional Effects on the Blood-Brain Barrier"

_ijms, 2021, doi:10.3390/ijms22041657_

Round 1
Reviewer 1 Report
In the work entitled: “Time-dependent internalization of polymer-coated silica nanoparticles in brain endothelial cells and morphological and functional effects on the blood-brain barrier”, Bittner and colleagues present a detailed study of the bionano interaction of nanoparticles with in vitro models of blood-brain barrier. The manuscript is nicely written and well organized, with a clear and wide introduction and deep discussion of the results. The work moves from biocompatibility studies of nanoparticles to their internalization in endothelial cells, to finish up with potential inflammation effects as well as effects on BBB permeability and formation. Finally, the authors present an interesting 3D model of BBB to further investigate NP exposure. One minor suggestion would be to add a schematic cartoon of the different systems used, including transwell monocultures and co-cultures and the 3D model. This will definitely help in quickly understand the set-up used along the many experiment presented.
Although the results look promising, there are some crucial aspects that need to be clarified before the manuscript can be taken into consideration for publication. In details, my main comments are:
- More emphasis should be given to the materials tested (although already published in ref 46. Being a work presenting nanoparticle biological effects, it would be of great help to have at least a table summarizing the physical-chemical properties of the nanoparticles, including size and z-potential. These information are somehow hidden in the Materials and Methods section and along the whole manuscript there is no mention at all of the size of the NP used.
- The authors nicely present the BBB formation (Appendix) and the effects of NP exposure of BBB integrity (by Papp and TEER). However, to fully exclude any disruption of the endothelial layer, it would be convenient to add confocal images of endothelial cells exposed to NP and stained with Claudin-5 and/or ZO-1 and/or VE-Cadherin (for AJs).
- In the discussion section, lines 350-351, authors claim that the tested NP were seen to not cross the BBB. However, there are not experimental evidences presented in the manuscript to justify the claim. I suggest to either remove the sentence or alternatively to perform transport studies of the fluorescent NPs with a transwell systems. Did the authors have the chance to check if NPs remain trapped in the transwell filter? (see Bramini et al., ACS Nano 2014).
- There are no images of pericytes. Although authors say (line 152-154) that in co-culture mode no NPs were found in pericytes and pericytes in monoculture instead were able to uptaken the NP, no data for such claims are presented. Please provide evidences, even in supp figures.
There are then some minor considerations the authors should address:
- In the results section, 2.1 (lines 102-103), please mention how the viability and toxicity have been evaluates (MTT and PI).
- In Figure 1, please change the X label “conc 1-3” with the real concentrations used in the studies. It would be much easier for the reader to go through the graphs.
- Add the n used for the statistics in each figure legend.
- Please, double check Figure numbering since there is no correspondence between the text and the real number of the Figures.
- Line 133 “data not shown”: please either remove the sentence related to structured illumination microscopy or show the data (even in a supplementary figure).
- Line 140, please add that the inverted epifluorescence microscope used for internalization studies is adopted with Apoptome (as far as I understood).
- In the section 2.3, authors refer more times to the “levels” of the proteins studies (i.e. line 161). Please, remove any quantification reference or provide quantification data (that could be mean fluorescence intensity, since fluorescence images are presented, or western blotting).
- Page 9, Figure 6 (not 4), please change the legends in a and b of the concentration with the real values used and add the legend that is missing in c.
- Line 433, please add the references the authors refer to.
- It would be nice to have in a supplementary figure some representative images of the cytotoxicity studies carried out with PI and Hoechst, to be able to correlate the images with the quantified data presented in the main text.
- Line 524-529: it is not clear when the Epifluorescence and the Confocal were used. Line 140 of the main text the authors say they used an epifluorescence for internalization study, that differ to what stated in the Material and Methods section. Please, could the authors clarify? In case of confocal images of internalized particles, it would be appropriate to present z-stack reconstruction.
- Could also the author better describe how the quantification of NP internalized was calculated? The % of internalize NP refers to the total amount of NP given to the cell? Was it calculated by fluorescent signal or number of NP?
Reviewer 2 Report
This work is devoted to the study of nanoparticles for the functional state of endothelial cells. The experiments were carried out on human brain endothelial cells (ECs) and blood-brain barrier models. The effect of nanoparticles on cell survival, expression of pro-inflammatory proteins, and blood-brain barrier permeability was investigated. The uptake of nanoparticles by endothelial cells was characterized. The work was done at a high technical level, but there are some comments, concerning the setting of experiments and style.
Comments.
Introduction.
Two paragraphs in the introduction contain an unnecessarily detailed presentation of the results obtained by the authors earlier (lines 57-83).
Results.
Figure 1 shows data on the effect of nanoparticles in the range of 7 orders of magnitude with only three concentrations. This does not make sense, since a 30-40% effect was observed only at maximum concentration. The concentration-dependence should be determined in a more narrow range.
Line 116. It is not clear why, to study the cytotoxic effect of nanoparticles, a concentration was chosen (2.49 x 10-3 μg/mL) at which they do not exhibit such an effect.
Line 118. The authors have no reason to assert that PCL- and PLLA-NPs had similar effects, since in fact there was no determination of concentration dependences for their action and their cytotoxic effects at the plateau levels were not evaluated.
Line 126. An error – it should be Figure 2. This figure actually duplicates the data shown in the previous figure. What is the reason to demonstrate for the second time that at 2.49 x 10-3 μg/mL concentration there is no effect of PCL-NPs?
Line 144. Please, change the number of the figure. Why the nanoparticles surrounded by a membrane are not shown in section b of the figure while in the text the authors are writing that they are there? It would be helpful to show a photo of the CD34 + EC. What is 100% in the picture? If this is the number of cells that have absorbed nanoparticles, then why is the designation uptake given on the ordinate? Complete explanation of statistical analysis of the data presented on this Figure (page 5) should be given.
Line 165. Legend to Figure 4. It is not correct to compare the effect of long-term incubation with TNF-α and IFN-γ (16 hours) and short-term incubation (2 hours) with PLLA-NPs. There was a strong signal of p-NF-κB in the cytoplasm in response to PLLA-NPs (Fig.4f). It might be that after 16 hours it will be seen in the nuclei. Why there were different positive controls for NF-κB and p-NF-κB staining? Please, indicate properly concentrations of both TNF-α and IFN-γ.
Page 7.The number of the Figure is wrong. There is no statistical processing of the data presented in the figure. The conclusions are not substantiated.
Figure 4 (the one on page 9, line 196) compares the effect of nanoparticles at concentrations of 2.49x10-7 and 2.49 μg / ml. It is not clear what is the point in investigating the effect of the homeopathic concentration of PCL- or PLLA-NPs.
The style of presentation in some places is rather heavy, the sentences are too long. The combination of the words medicine and nanomedicine in one sentence next to each other (lines 39-40) sounds annoying.
Lines 118-119. “the results from the viability assay showed similar results”. The style should be corrected in some places.
Round 2
Reviewer 1 Report
The authors fully addressed all the concerns. The work is ready for publication as it is.
Reviewer 2 Report
The authors took into account the comments/ In the present form the manuscript can be accepted.